# Small Cell Lung Cancer: Emerging Targets and Strategies for Precision Therapy

**DOI:** 10.3390/cancers15164016

**Published:** 2023-08-08

**Authors:** Shruti R. Patel, Millie Das

**Affiliations:** 1Department of Medicine, Division of Medical Oncology, Stanford Cancer Institute, Stanford University, Palo Alto, CA 94305, USA; shrutipatel@stanford.edu; 2Department of Medicine, Oncology Section, VA Palo Alto Health Care System, Palo Alto, CA 94304, USA

**Keywords:** small cell, targeted therapy, PARP, DLL3, epigenetic modulators, ADC

## Abstract

**Simple Summary:**

Small cell lung cancer (SCLC) remains a difficult-to-treat disease and is associated with a poor prognosis in most patients. Discovering biomarkers and developing targeted treatment options for patients with SCLC has been challenging though there are emerging therapies that are showing promise. A better understanding of the underlying biology and specific molecular targets in SCLC has led to newer drug strategies and combinations that can hopefully lead to better outcomes for our patients.

**Abstract:**

Small cell lung cancer is an aggressive subtype of lung cancer with limited treatment options. Precision medicine has revolutionized cancer treatment for many tumor types but progress in SCLC has been slower due to the lack of targetable biomarkers. This review article provides an overview of emerging strategies for precision therapy in SCLC. Targeted therapies include targeted kinase inhibitors, monoclonal antibodies, angiogenesis inhibitors, antibody–drug conjugates, PARP inhibitors, and epigenetic modulators. Angiogenesis inhibitors and DNA-damaging agents, such as PARP and ATR inhibitors, have been explored in SCLC with limited success to date although trials are ongoing. The potential of targeting DLL3, a NOTCH ligand, through antibody–drug conjugates, bispecific T-cell engagers, and CAR T-cell therapy, has opened up new therapeutic options moving forward. Additionally, new research in epigenetic therapeutics in reversing transcriptional repression, modulating anti-tumor immunity, and utilizing antibody–drug conjugates to target cell surface-specific targets in SCLC are also being investigated. While progress in precision therapy for SCLC has been challenging, recent advancements provide optimism for improved treatment outcomes. However, several challenges remain and will need to be addressed, including drug resistance and tumor heterogeneity. Further research and biomarker-selected clinical trials are necessary to develop effective precision therapies for SCLC patients.

## 1. Introduction

Small cell lung cancer (SCLC) is a subtype of lung cancer that is highly aggressive and represents approximately 15% of all lung cancer cases [1,2,3]. Despite the initial responsiveness to chemotherapy, the majority of patients with SCLC will eventually relapse, with an overall 5-year survival rate of <7% [4]. Most SCLC cases are associated with exposure to tobacco carcinogens with a median age at diagnosis of 70 years old [5]. Only ~2% of SCLC cases arise in never-smokers (defined as lifetime smoking of fewer than 100 cigarettes) [6].

SCLC remains a major clinical challenge mainly due to difficulties in early detection and the lack of effective therapeutic options. Unfortunately, most SCLC patients are diagnosed at an advanced stage with a treatment goal of palliation [5]. Typically, platinum-based alkylating agents (such as cisplatin/carboplatin) and topoisomerase inhibitors (such as irinotecan/etoposide) are used in combination as a first-line treatment and now also with the addition of a PD-L1 inhibitor (IMpower133 and CASPIAN) [5,7]. Unfortunately, despite the recent advance of adding immunotherapy in the frontline setting in patients with extensive stage SCLC (ES-SCLC), outcomes remain poor for these patients, with the vast majority of patients progressing while on maintenance immunotherapy [8]. Back in 1998, the topoisomerase I inhibitor, topotecan, was approved by the US Food and Drug Administration (FDA) for recurrent SCLC, though topotecan had demonstrated modest clinical efficacy with an overall response rate (ORR) in the 20% range [9,10]. Topotecan remained the only FDA-approved drug in relapsed SCLC until only recently when lurbinectedin was granted accelerated FDA approval in June 2020 based upon an ORR = 35% in 105 patients in a phase II basket trial [11]. The confirmatory phase III LAGOON trial is underway, comparing lurbinectedin alone or in combination with irinotecan versus topotecan or irinotecan in patients with relapsed SCLC [12]. Given the limited advances and options over the past few decades, there remains an urgent need for new therapeutic approaches to improve outcomes for SCLC patients.

In recent years, the emergence of precision medicine has revolutionized cancer treatment by providing a more individualized approach to therapy. Precision medicine involves the identification of specific molecular targets that are critical for tumor growth and survival and the development of drugs that specifically target these gene alterations. This approach has been highly successful in several cancer types, including non-small cell lung cancer (NSCLC), prostate cancer, ovarian cancer, and cholangiocarcinoma [13]. However, progress in SCLC has been slower due to the lack of targetable biomarkers or molecular pathways that appear to drive SCLC tumorigenesis.

While the pace of discovery in SCLC has not matched NSCLC, there has been significant recent progress in the understanding of the molecular mechanisms that underlie SCLC tumorigenesis which may lead to the identification of potential targets for precision therapy. Therapies that specifically target oncogenic drivers have the potential to provide significant benefits in terms of improved response rates, prolonged progression-free survival (PFS), and reduced toxicity compared to traditional cytotoxic chemotherapy. Key molecular pathways that are critical for SCLC growth and survival have been studied extensively but genomic profiling has not identified obvious mutationally defined subtypes of SCLC. Some of the molecular alterations seen in SCLC include those in the pathways related to the oncogene MYC, the tumor suppressor genes RB1, TP53, and PTEN, NOTCH, and the chromatin regulator CREBBP41 [5].

Targeted therapies fall into several categories, including targeted kinase inhibitors (TKIs), monoclonal antibodies, angiogenesis inhibitors, poly (ADP-ribose) polymerase (PARP) inhibitors, antibody–drug conjugates (ADCs), and epigenetic modulators. Much of the research in targeted therapies in SCLC has focused on targeted kinase inhibitors, drugs that specifically inhibit the activity of oncogenic kinases that are critical for tumor growth and survival. 

The objective of this review article is to provide an overview of the emerging targets and strategies for precision therapy in SCLC. We will discuss the current understanding of the molecular mechanisms that underlie SCLC tumorigenesis, the importance of targeted therapy in treating SCLC, and the latest developments in precision medicine for SCLC. We will also highlight the challenges that remain in the development of effective precision therapies for SCLC, including drug resistance and tumor heterogeneity.

## 2. Oncogenic Drivers and Targeted Therapy in SCLC 

As described, SCLC is a highly aggressive and lethal malignancy which further underscores the need for better understanding of the underlying molecular mechanisms driving its pathogenesis. The identification of oncogenic drivers in SCLC has provided a significant impetus for the development of targeted therapies. This section will provide an overview of the known oncogenic drivers in SCLC, the mechanisms of their activation, and the status of targeted therapies for these molecular drivers.

The pathogenesis of SCLC is complex and involves the activation of multiple oncogenic drivers. Some of the well-known oncogenic drivers in SCLC include MYC, TP53, RB1, PTEN, and NOTCH1. MYC amplification is one of the most frequent genetic abnormalities in SCLC and promotes cell proliferation, survival, and metastasis [14]. TP53 and PTEN mutations, which lead to loss of tumor suppressor function, are also common in SCLC and are associated with poor prognosis [15]. Additionally, loss of function mutations in RB1 and PTEN are prevalent in SCLC and lead to the dysregulation of multiple signaling pathways, including PI3K/AKT/mTOR, RAS/RAF/MEK/ERK, and NOTCH1 signaling, contributing to the aggressive nature of this disease. 

Despite the identification of multiple oncogenic drivers in SCLC over the past two decades, there are limited targeted therapies that are available and effective for these drivers. The therapies that have been studied encompass a wide range of agents, including vascular endothelial growth factor (VEGF) inhibitors, PARP and ataxia telangiectasia and Rad3-related protein (ATR) inhibitors, and mammalian target of rapamycin (mTOR) inhibitors. Unfortunately, despite these efforts, none of these targeted agents have demonstrated a survival benefit in SCLC. Additionally, clinical trials evaluating antiangiogenic agents such as bevacizumab, thalidomide, and sorafenib have yielded disappointing results, failing to improve overall survival in SCLC patients. 

### 2.1. Antiangiogenic Agents

Among the angiogenesis inhibitors, an antibody that targets VEGF, bevacizumab, has been the most extensively investigated. In two phase 2 trials focused on the maintenance setting in limited-stage SCLC patients, initial findings favored those patients receiving maintenance bevacizumab after the completion of definitive chemoradiation in terms of response rates and PFS [16]. However, subsequent trials revealed an increased risk of tracheoesophageal fistula development when bevacizumab was used in this context [17]. In studies examining bevacizumab in combination with platinum-based chemotherapy in extensive-stage SCLC, an improvement in PFS was observed in a highly select patient population but no significant improvement in median overall survival (OS) was demonstrated [18]. Similarly, a study exploring the combination of bevacizumab and paclitaxel in chemo-sensitive ES-SCLC failed to yield any noteworthy clinical outcomes [19]. 

The VEGF TKI, sunitinib, was studied in the phase II placebo-controlled trial CALGB 30504 which investigated the role of maintenance sunitinib in ES-SCLC. Sunitinib administration after platinum-based chemotherapy with etoposide exhibited only a modest increase in median PFS [20]. Aflibercept is an angiogenesis inhibitor with a unique mode of action, binding to both VEGF-A and VEGF-B, thereby preventing their binding to cell receptors. A phase II trial investigated the use of topotecan with or without aflibercept in platinum-treated ES-SCLC patients [21]. The trial revealed an improvement in 3-month PFS in platinum-refractory disease but not in platinum-sensitive disease, with no improvement noted in OS in either group.

The failure of antiangiogenic agents to improve OS in SCLC further underscores the fundamental biological differences between SCLC and NSCLC, where bevacizumab has previously been shown to improve survival [22]. SCLC has proven to be a daunting challenge in terms of drug development, primarily due to its high number of somatic mutations and inherent heterogeneity. Consequently, finding effective targeted therapies for SCLC remains an ongoing struggle.

### 2.2. DNA-Damaging Agents

Historically, high tumor mutational burden (TMB) has been considered to be a potential predictive factor for immunotherapy response [23]. Unfortunately, despite the fact that SCLC often has high TMB, the immunotherapy response has been limited due to significant immunosuppression in the tumor microenvironment characterized by limited T-cell infiltration and impaired antigen presentation [24,25]. This finding may explain why clinical trials evaluating the efficacy of the PD-1/PD-L1 blockade in SCLC patients have shown only modest improvements in the overall response rate and survival [26]. DNA damaging agents have shown significant potential as therapeutic targets in SCLC with inhibitors targeting DNA damage repair proteins showing promise in preclinical models. These inhibitors are currently being investigated in clinical trials, both as single agents and in combination with other therapies. Although primarily recognized for its role in repairing DNA damage and regulating the cell cycle, the DNA damage repair pathway has also been found to influence the antitumor immune response [27]. 

Several drugs have been developed to target DNA damage repair (DDR) proteins, including inhibitors of PARP, ATR, and checkpoint kinase 1 (CHK1). Some DDR inhibitors are approved in other tumor types including breast, ovarian, and prostate cancers. To date, a limited number of trials have included SCLC patients but the available data indicate the potentially promising activity of PARP inhibitors in relapsed SCLC patients (Table 1). Multiple combination trials of PARP inhibitors with temozolomide in patients with relapsed SCLC have yielded encouraging overall response rates in the 40% range [28,29,30]. Additionally, an emerging candidate biomarker, Schlafen 11 (SLFN11), may hold the potential to identify the subsets of SCLC patients who are most susceptible to PARP inhibition. The presence of SLFN11 correlates with sensitivity to DDR and PARP inhibitors while low or absent expression is associated with resistance [31]. Efforts are currently underway to clinically validate SLFN11 in SCLC. In a phase 2 trial comparing the PARP inhibitor veliparib plus temozolomide to placebo/temozolomide in relapsed SCLC patients, SLFN-11 expression was detected in approximately 50% of tumors using immunohistochemistry (IHC). Moreover, the investigators noted that the SLFN11-positive cohort of patients receiving PARP inhibition had significantly prolonged PFS and OS compared to the SLFN11-negative group of patients [28]. More recently, the phase II randomized SWOG 1929 study conducted in patients with SLFN11 positive ES-SCLC found that maintenance of atezolizumab treatment plus the PARP inhibitor, talazoparib, improved PFS in SLFN-11 selected patients compared to maintenance of atezolizumab treatment alone. Importantly, this study also demonstrated the feasibility of conducting biomarker selected trials in SCLC [32].

In SCLC, the effectiveness of PARP inhibition also appears to be dependent on high levels of replication stress and genomic instability. Single-agent talazoparib was tested in an expansion cohort of platinum-sensitive patients who had relapsed, with two out of the 23 patients in the SCLC cohort achieving partial responses lasting 12.0 and 15.3 weeks, respectively, and four patients experiencing stable disease lasting over 16 weeks. The median PFS for the overall SCLC cohort was 11.1 weeks [34]. In newly diagnosed and treatment-naïve patients, veliparib combined with cisplatin and etoposide demonstrated a PFS of 6.1 months in the experimental group, compared to 5.5 months in the control arm (unstratified HR 0.75; one-sided *p* = 0.06) [35]. There was a trend towards improved OS in the experimental group although this difference did not reach statistical significance [35].

Preclinical observations suggest that ATR inhibitors may be particularly effective in TP53 or ATM-deficient models. As ATM-deficient tumors represent about 8% of SCLC cases and nearly all SCLC cases have TP53 mutations, phase 1 trials with ATR inhibitors have included SCLC patients among other solid tumors. Results from these studies are pending and will inform further trials in SCLC [36].

CHK1, a vital serine–threonine protein kinase, acts as a principal mediator of cell-cycle arrest in response to DNA damage in cells lacking functional TP53 [37]. CHK1 inhibitors have been shown to enhance the effects of DNA-damaging treatments in triple-negative breast cancers and head and neck cancers, where TP53 mutations are frequently observed [38]. One study demonstrated the significant efficacy of LY2606368, prexasertib, as a standalone agent in SCLC models in vitro and in vivo. Prexasertib was found to improve the effects of cisplatin or the PARP inhibitor, olaparib, and improved the response in platinum-resistant models. Proteomic analysis of these models revealed that CHK1 and MYC emerged as top predictive biomarkers for sensitivity to prexasertib. These findings suggested that CHK1 inhibition may hold particular promise in SCLC cases characterized by MYC amplification or overexpression of MYC protein [39].

However, a subsequent phase II study of prexasertib in SCLC patients who had progressed after one or two prior therapies noted disappointing results, with an ORR = 5.2% in platinum-sensitive patients and an ORR = 0% in platinum-resistant/platinum-refractory patients [40].

### 2.3. DLL3 Trials

NOTCH1 mutations lead to the constitutive activation of its signaling pathway, promoting cell proliferation, migration, and invasion [41]. NOTCH pathways have been demonstrated to be both tumor suppressive and pro-tumorigenic in SCLC [42]. Delta-like ligand 3 (DLL3), an inhibitory ligand of the NOTCH pathway, exhibits aberrant expression on the cell surface of approximately 85% of SCLC cells while showing minimal expression in normal tissues [43,44]. Experimental models of SCLC conducted in vitro have provided evidence suggesting the involvement of DLL3 in facilitating tumor growth, migration, and invasion [45]. The unique expression profile of DLL3 has paved the way for the development of therapeutics specifically targeting DLL3 in SCLC [46] (Table 2).

A novel approach to targeting DLL3-expressing tumor cells has been developed in the form of BiTE molecules and chimeric antigen receptor (CAR) T cells which function by utilizing immunotherapy to harness T-cell cytotoxicity. Bispecific DLL3/CD3 IgG-like T-cell engagers induce the formation of MHC-independent cytolytic synapses by binding to DLL3 on tumor cells and CD3 on T cells, thereby initiating T cell-mediated antitumor cytolytic activity [50]. One such agent, tarlatamab, a bispecific T-cell engager, has advanced the furthest in clinical trials with a phase 1 study of tarlatamab in SCLC patients reporting an impressive median OS of 13.2 months in a predominantly third-line patient population [51]. The ORR was 23.4% and the DCR was 51.4%. Encouragingly, among the responders, there were two patients who achieved a complete response and the median duration of the response (DOR) was 13.1 months, indicating the potential for durable and longer-term responses in patients with relapsed disease. The study also highlighted toxicities with BiTE therapy, such as cytokine release syndrome (CRS) and immune effector cell-associated neurotoxicity syndrome (ICANS). Other DLL3 T-cell engagers, including HPN328 and BI764532, have shown promising results, with HPN328 demonstrating positive outcomes in 6 of 15 patients (40%) achieving a decrease in the sum of target lesion diameters in a study presented at ASCO 2022 [48] and BI764532 achieving a 33% ORR in 24 SCLC patients, as reported at ASCO 2023 [49].

AMG 119 is a CAR T approach that represents an adoptive cellular therapy approach involving genetically modified autologous T cells that express a CAR targeting DLL3. Unlike tarlatamab, AMG 119 CAR T cells have the potential for prolonged antitumor activity with a single administration. A phase 1 study of AMG 119 in SCLC has reported results in four evaluable patients, with one patient achieving a confirmed PR 1.1 months after the first dose and two patients with SD, including a patient who experienced a 16% decrease in the sum of the target lesions from baseline [52]. Enrollment onto this trial is currently paused but hopefully will resume in the future. 

### 2.4. Epigenetic Therapeutics

Epigenetic reprogramming has emerged as a promising therapeutic target in SCLC with several recent studies highlighting the involvement of epigenetic modifications, such as DNA methylation, chromatin accessibility, and histone modifications, in the pathogenesis of this aggressive disease [53]. The ability to alter transcriptional programming may be effective in SCLC by reversing transcriptional repression of MHC1. Additionally, epigenetic programming alterations can also modulate anti-tumor immunity by reversing T-cell dysfunction and improving T-cell trafficking.

The enhancer of zeste homolog 2 (EZH2), an enzymatic subunit of polycomb repressive complex 2 (PRC2), is a transcription regulator and is highly expressed in SCLC compared to normal lung tissues. It is associated with transcription repression and aberrant methylation and is implicated in multiple functions to promote tumorigenesis, including cell cycle regulation and the down-regulation of MHC1 and C–C motif chemokine ligand 5 (CCL5) expression [54]. Histone modification has been implicated in the development of acquired chemoresistance in patients based on the EZH2-SLFN11 axis. The EZH2-SLFN11 axis has been found to regulate sensitivity to chemotherapy in SCLC; elevated expression of SLFN11 enhances sensitivity to chemotherapy. Preclinical investigations employing an EZH2 inhibitor in combination with standard cytotoxic therapies have been shown to prevent the emergence of chemoresistance and to enhance the response to chemotherapy in SCLC animal models [53].

Another histone-modifying enzyme, lysine demethylase 1 (LSD1 or KDM1A), has been implicated as a potential therapeutic target in SCLC. LSD1 inhibition leads to the activation of NOTCH and suppression of ASCL1 and increases MHC1 expression [55]. When combined with PD-1 inhibitors, the LSD1 blockade enhanced the sensitivity of SCLC tumors to immunotherapy [56]. The LSD1-targeted drug GSK2879552 has demonstrated effective antitumor activity in SCLC cell lines and primary samples. However, a clinical trial evaluating GSK2879552 in relapsed/refractory SCLC was terminated due to poor disease control rates and high rates of adverse events [57]. Future investigations will focus on clinical trials combining epigenetic drugs with the checkpoint immune blockade and/or chemotherapy to explore their synergistic effects and therapeutic potential in SCLC. A phase 1/2 study of the selective inhibitor of LSD1, iadaemstat, plus PD-1 inhibitors as maintenance therapy in SCLC who complete induction chemoimmunotherapy is supported by the NCI-CTEP and will be opened. 

### 2.5. Novel Antibody–Drug Conjugates

Antibody–drug conjugates represent the latest advance within precision oncology through the targeting of cell surface-specific targets in various malignancies, including in SCLC (Table 3). ADCs are made up of three components: (1) a monoclonal antibody that selectively binds an antigen on the tumor cell surface, (2) a cytotoxic drug payload, and (3) a linker that connects the two [58]. Through this unique mechanism of action, ADCs can specifically target tumor cells and become internalized to fuse with lysosomes which leads to the release of the cytotoxic agent to induce cell death and minimize off-target effects. 

Trophoblast cell surface antigen (TROP2) is expressed in many epithelial cancers and overexpression is associated with poor survival. A high expression of TROP 2 has been noted in 18% of high-grade neuroendocrine tumors, including 10% SCLC [62]. Sacituzumab govitecan (IMMU-132) is a first-in-class anti-TROP2 ADC that consists of an anti-TROP2 monoclonal antibody linked to the topoisomerase 1 inhibitor SN-38. This ADC has so far been evaluated in 62 pre-treated SCLC patients, indicating an ORR of 18%, median DOR of 5.7 months, PFS of 3.7 months, and OS of 7.1 months [59]. The ongoing TROPiCS-03 is a phase II trial that is further investigating sacituzumab govitecan in patients with metastatic solid tumors including SCLC [63].

Ifinatamab deruxtecan (I-DXd) is a novel ADC consisting of an anti-B7-H3 antibody linked with Dxd, a DNA topoisomerase inhibiting anti-tumor agent. I-DXd has shown promising durable tumor response in patients with several types of heavily pretreated cancers including lung, prostate, and esophageal cancer. Among 19 SCLC patients, 53% had confirmed responses with a median DOR of 5.5 months [60]. A phase 2 study is ongoing with I-DXd in patients with pretreated ES-SCLC [64].

Seizure related 6 homolog (SEZ6) is a cell surface protein that has been found to be highly expressed in neuroendocrine tumors with minimal expression in the majority of normal tissues which makes it another potential target for SCLC [65]. ABBV-011 is an ADC that combines an anti-SEZ6 antibody with calicheamicin, a potent cytotoxic agent. Calicheamicin has been used in two other approved ADCs for hematologic malignancies [66]. A recent phase 1 trial with patients with relapsed/refractory SCLC demonstrated favorable tolerability and exhibited significant antitumor activity with an overall response rate of 25% and a duration of response reaching 4.2 months [61].

Rovalpituzumab tesirine, an ADC targeting DLL3, had shown promise in preclinical models of SCLC and large-cell neuroendocrine carcinoma. It consists of a humanized DLL3-specific IgG1 monoclonal antibody linked to the DNA cross-linking agent pyrolobenzodiazepine (PDB) via a protease-cleavable linker [46]. In a phase 1 study, rovalpituzumab tesirine demonstrated durable responses with a subsequent phase 2 trial (TRINITY) in DLL3-expressing SCLC patients reporting a median OS of 5.6 months [67]. However, the phase 3 trial (TAHOE) comparing rovalpituzumab tesirine to topotecan as second-line therapy for SCLC was discontinued based on interim analysis demonstrating shorter OS in the rovalpituzumab tesirine arm which was felt to be related to the significant toxicities of the ADC [47]. Though further clinical development of this particular DLL3 targeting ADC has been discontinued, DLL3 has remained a viable therapeutic target of interest in SCLC.

### 2.6. Other Possible Therapies

Fucosyl-GM1 (FucGM1) is a monosialoganglioside highly expressed on the surface of SCLC cells but it has limited expression in normal tissues [68]. BMS-986012 is a nonfucosylated, first-in-class, and fully human immunoglobulin G1 monoclonal antibody that binds to FucGM1. BMS-986012 was found to have significant efficacy in SCLC mouse xenografts and was evaluated in humans in a phase 1 study that demonstrated limited activity as a monotherapy (ORR 4%, *n* = 77). When the agent was combined with the anti-PD-1 antibody nivolumab, the ORR increased to 38% (*n* = 29). A study combining BMS-986012 with chemotherapy and nivolumab is currently underway in the first-line setting for ES-SCLC [69].

THZ1 is a covalent inhibitor of cyclin-dependent kinase 7 (CDK7) which has demonstrated promising anti-tumor activity against various cancer types. CDK7 is a cell cycle regulator that promotes cell cycle progression through the inhibition of transcription rather than through direct phosphorylation of classical CDK targets as seen in breast cancer [70]. Using a high throughput cellular screen of a diverse chemical library, investigators noted that SCLC is sensitive to transcription-targeting drugs, specifically to THZ1, representing a promising new drug candidate for the treatment of SCLC [71].

Finally, there has been interest in exploring CD47 (integrin-associated protein) as a potential therapeutic target in various malignancies, including in SCLC. CD47 is a cell surface ligand that is overexpressed by different cancer types and plays an essential role in the immune response by sending a “don’t eat me” signal to prevent phagocytosis by macrophages. Pre-clinical studies have examined the interplay between radiation and anti-CD47 therapy in SCLC mouse models as radiation therapy is commonly used in conjunction with systemic therapies for both palliative and curative purposes in SCLC [72]. In mice with SCLC tumors treated with the anti-CD47 blockade along with radiation, not only was there a greater anti-tumor effect compared to radiation alone but tumors outside the field of radiation were also reduced in size and these mice developed more resistance to re-challenge newly-injected SCLC cells. These preclinical findings lend support to the investigation of the combination of anti-CD47 therapy with radiation in SCLC [73].

### 2.7. Characterization of SCLC Subtypes

In addition to the work being conducted to explore targeted therapies in SCLC, there is extensive work being performed to further characterize these tumors based on the relative expression levels of key transcription factors, namely achaete-scute homolog 1 (ASCL1), neurogenic differentiation factor 1 (NEUROD1), yes-associated protein 1 (YAP1), and POU class 2 homeobox 3 (POU2F3) [74]. Immunohistochemical analysis revealed challenges in confirming the SCLC-Y subtype due to the low protein expression level of YAP1 in SCLC tissues and as a result, a more updated classification system was created. In addition to SCLC-A, SCLC-N, and SCLC-P, a new subtype was found, SCLC-I, which represents a subtype with low expression levels of ASCL1, NEUROD1, and POU2F3 [75]. At this point, there are substantial preclinical data available on the therapeutic implications of these four subtypes. Further studies are critical in evaluating the clinicopathological features, immunity profiles, and treatment outcomes of these four subtypes. Understanding the distinct characteristics and biological behaviors of these subtypes may aid in the development of targeted therapies tailored to each subtype.

## 3. Conclusions

In conclusion, SCLC continues to pose a significant therapeutic challenge for patients with this disease. While precision medicine has improved survival in many cancer types, progress in SCLC has been slower due to the absence of actionable biomarkers and molecular pathways that drive SCLC tumorigenesis. However, recent advancements in the understanding of the molecular transcriptional subtypes of SCLC and of the molecular mechanisms underlying SCLC tumorigenesis have provided hope for the identification of potential targets for precision therapy and more personalized treatment options for patients. 

To date, most targeted agents have failed to demonstrate a survival benefit in clinical trials as outlined above, highlighting the challenges in developing more effective therapies for SCLC. Future directions should focus on developing combination strategies to overcome resistance mechanisms and tumor heterogeneity and to more effectively employ the immune system’s ability to attack cancer cells. Using precision medicine, it is also critical to provide clinical validation of potential biomarkers in SCLC and to design prospective biomarker-selected clinical trials to provide much-needed improvements in patient outcomes. 

## Figures and Tables

**Table 1 cancers-15-04016-t001:** Second-line PARP inhibitor–Temozolomide combinations compared to Topotecan.

	Topotecan [10]	Talazoparib + TMZ [30]	Olaparib + TMZ [33]	Veliparib + TMZ [28]
*n*	151	28	50	104
ORR	21.9%	39.3%	41.7%	39%
Median DOR	6.4 mo	4.3 mo	4.3 mo	4.6 mo
Median PFS	ND	4.3 mo	4.2 mo	3.8 mo
Median OS	8.75 mo	11.9 mo	8.5 mo	8.2 mo

**Table 2 cancers-15-04016-t002:** DLL3 targeted agents.

	RovaT [47]	Tarlatamab	HPN328 [48]	BI76453 [49]
*n*	287	106	11	57
ORR	14.3%	23%	27%	51%
Median DOR	3.5 mo	13 mo	ND	ND
Median PFS	3 mo	3.7 mo	ND	ND
Median OS	6.3 mo	13.2 mo	ND	ND

**Table 3 cancers-15-04016-t003:** ADCs in SCLC.

	Sacituzumab Govitecan [59]	Ifinatamab Deruxtecan [60]	ABBV-011 [61]	RovaT [47]
Tumor antigen target	TROP2	B7-H3	SEZ6	DLL3
Cytotoxic agent	SN-38	Dxd	Calicheamicin	Pyrrolobenzodiazepine
*n*	62	19	40	287
ORR	18%	53%	25%	14.3%
Median DOR	5.7 mo	5.5 mo	4.2 mo	3.5 mo
Median PFS	3.7 mo	ND	ND	3 mo
Median OS	7.1 mo	ND	ND	6.3 mo

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
