# Peer review of "Small Cell Lung Cancer: Emerging Targets and Strategies for Precision Therapy"

_cancers, 2023, doi:10.3390/cancers15164016_

Round 1

Reviewer 1 Report

In this review, Patel and Das discuss therapeutic strategies to treat small cell lung cancer. After a general introduction on SCLC and its response to chemotherapy, the authors focus on several approaches of targeted therapy, providing a broad overview of the different clinical trials related to each of these strategies. The manuscript is comprehensive and very well written. Some suggestions that may improve the paper are listed below:

1. It has been shown that the expression of certain transcription factors can be used to classify SCLC in subtypes displaying different vulnerabilities to treatment (Gay et al., Cancer Cell 2021). This study should be discussed in the manuscript.

2. The paragraph on rovalpituzumab tesirine (2.3) should be moved to the section describing antibody drug conjugates (paragraph 2.5).

3. I would change the title of the second section of the manuscript, since several of the outlined strategies are not designed to specifically target oncogenic drivers of SCLC.

4. It would be helpful to include one or two figures illustrating the different strategies discussed in the manuscript.

Author Response

Response to Reviewer 1 Comments

Point 1: In this review, Patel and Das discuss therapeutic strategies to treat small cell lung cancer. After a general introduction on SCLC and its response to chemotherapy, the authors focus on several approaches of targeted therapy, providing a broad overview of the different clinical trials related to each of these strategies. The manuscript is comprehensive and very well written.

Response 1: Thank you for your review and these comments.

Point 2: It has been shown that the expression of certain transcription factors can be used to classify SCLC in subtypes displaying different vulnerabilities to treatment (Gay et al., Cancer Cell 2021). This study should be discussed in the manuscript.

Response 2: Thank you for this suggestion. We agree with your recommendation and have added the following paragraph to the manuscript in a new section titled “2.7 Characterization of SCLC subtypes”

In addition to the work being doing to explore targeted therapies in SCLC, there is extensive work being done to further characterize these tumors based on the relative expression levels of key transcription factors, namely achaete-scute homolog 1 (ASCL1), neurogenic differentiation factor 1 (NEUROD1), yes-associated protein 1 (YAP1), and POU class 2 homeobox 3 (POU2F3). Immunohistochemical analysis revealed challenges in confirming the SCLC-Y subtype due to the low protein expression level of YAP1 in SCLC tissues and as a result, a more updated classification system was created. In addition to SCLC-A, SCLC-N, and SCLC-P, a new subtype was found, SCLC-I, which represents a represents a subtype with low expression levels of ASCL1, NEUROD1, and POU2F3. At this point, there is substantial preclinical data available on the therapeutic implications of these four subtypes. Further studies are critical in evaluating the clinicopathological features, immunity profiles, and treatment outcomes of these four subtypes. Understanding the distinct characteristics and biological behaviors of these subtypes may aid in the development of targeted therapies tailored to each subtype.

Point 3: The paragraph on rovalpituzumab tesirine (2.3) should be moved to the section describing antibody drug conjugates (paragraph 2.5).

Response 3: We have moved Rova-T to section 2.5 on Page 6 and have also added it to Table 3 on Page 9.

Point 4:  I would change the title of the second section of the manuscript, since several of the outlined strategies are not designed to specifically target oncogenic drivers of SCLC.

Response 4: We have changed that section’s title to “Oncogenic Drivers and Targeted Therapies in SCLC”

Point 5:  It would be helpful to include one or two figures illustrating the different strategies discussed in the manuscript.

Response 5: Thank you for this comment. We do not have the ability to create a figure, but are happy to reach out to a colleague to obtain the copywrite for a figure illustrating one of the different strategies discussed in the manuscript. As this will likely delay publication, please let us know if you think this is critical to the manuscript.

Reviewer 2 Report

There are limited information on targeted molecules and therapies for SCLC. This review should provide a better understanding of  those targets and strategies.

Author Response

Point 1: There are limited information on targeted molecules and therapies for SCLC. This review should provide a better understanding of those targets and strategies.

Response 1: Thank you for your review and these comments.